# Secretory Leukocyte Protease Inhibitor: A Pleiotropic Molecule for the Potential Diagnosis of and Therapy for Acute Kidney Injury

**DOI:** 10.3390/ijms262311631

**Published:** 2025-11-30

**Authors:** Rui Chen, Shiyun Gu, Fenfen Xiong, Lili Ji, Zhi-Jun Zhang, Bin Yang, Yuanyuan Wu

**Affiliations:** 1Department of Pathology, Medical School of Nantong University, Nantong 226001, China; 2331310038@stmail.ntu.edu.cn (R.C.); liliji79@ntu.edu.cn (L.J.); 2Nantong-Leicester Joint Institute of Kidney Science, Department of Nephrology, Affiliated Hospital of Nantong University, Medical School of Nantong University, Nantong 226001, China; 3Department of Human Anatomy, Medicine School of Nantong University, Nantong 226001, China; zhzhj@ntu.edu.cn; 4Renal Research Group, Cardiovascular Sciences, School of Medical Sciences, University of Leicester, University Hospitals of Leicester, Leicester LE1 9HN, UK

**Keywords:** acute kidney injury, biomarker, immunomodulation, SLPI, therapy

## Abstract

Acute kidney injury (AKI) remains a common clinical syndrome associated with high morbidity and mortality. However, effective diagnostic biomarkers and specific therapeutic interventions are still lacking. Secretory leukocyte protease inhibitor (SLPI), a serine protease inhibitor with pleiotropic functions, has emerged as an early diagnostic and prognostic biomarker for AKI. Clinical studies reveal significant elevation of serum SLPI in AKI patients compared to non-AKI patients at the acute phase following post-cardiovascular surgery, supporting its diagnostic potential. Furthermore, evidence also suggests that SLPI showed prognostic value for kidney transplantation and chronic kidney disease progression associated with diverse etiology, including diabetes. In addition, current evidence highlights the biological functions of SLPI in inhibiting NF-κB activities, suppressing neutrophil extracellular trap formation, modulating phagocytosis, regulating cell apoptosis, proliferation, differentiation, and potentially fibrosis across various disease contexts. Preclinical studies demonstrate that administration of recombinant SLPI ameliorates renal dysfunction in multiple AKI models, including ischemia–reperfusion injury and nephrotoxic models induced by gentamicin or cisplatin. Furthermore, the antifibrotic properties of SLPI underscore its therapeutic potential in halting AKI progression to chronic kidney disease. By integrating available evidence, this review aims to elucidate that, as an early acute-phase response molecule, SLPI serves dual roles as not only an early diagnostic and prognostic biomarker for AKI, but also a renoprotective molecule countering kidney injury.

## 1. Introduction

Acute kidney injury (AKI) describes a rapid deterioration of renal function with an incidence of 10–15% in hospitalized patients and exceeding 50% in the intensive care unit (ICU) [1]. AKI survivors are at significantly increased risk of developing chronic kidney disease (CKD) and of progressing to end-stage renal disease, with risks ninefold and threefold higher, respectively, compared with non-AKI patients [2,3]. KDIGO (Kidney Disease: Improving Global Outcome) criteria are widely used for clinical AKI diagnosis and mainly rely on detecting a significant elevation in serum creatinine. However, this elevation typically occurs only after renal function has declined by over 50%, potentially missing the critical window of early diagnosis and treatment [4]. Effective new biomarkers are urgently needed for the timely recognition of AKI, monitoring the progression of CKD, and guiding mechanism-based therapies.

Current soluble biomarkers in blood and urine for early AKI diagnosis have largely focused on indicators of renal parenchymal damage and sterile inflammation, such as kidney injury molecule-1 (KIM-1) [5,6], neutrophil gelatinase-associated lipocalin (NGAL) [7], liver-type fatty acid-binding protein [8], and interleukin (IL)-18 [9]. However, given the diversity and complexity of clinical AKI, more sensitive and specific biomarkers are needed to improve early prediction and enable timely therapeutic intervention, thereby reducing AKI incidence and improving outcomes. Secretory leukocyte protease inhibitor (SLPI) is a small protein, approximately 11.7 kDa, initially identified in the mucosal secretions, and it is produced mainly by inflammatory cells and epithelial cells [10,11,12,13,14]. Elevated circulating SLPI has been identified as an early diagnostic biomarker of AKI following cardiovascular surgery involving cardiopulmonary bypass, a procedure often associated with kidney ischemia–reperfusion (IR) injury [15]. Our previous work indicated that SLPI was significantly upregulated in kidneys from deceased donors compared to in living donors at both 30 min after transplantation [16], suggesting it as an acute-phase-responsive molecule. These findings suggest the clinical utility of SLPI in AKI risk stratification and early diagnosis.

The classical antiprotease activity of SLPI protects the host epithelia from excessive proteolytic damage by inhibiting proteases released by abundant infiltrated immune cells at the injury site. By acting on mucosal surfaces, SLPI also contributes the first-line defense against microbial infections [17]. Beyond these roles, SLPI has been shown to inhibit nuclear factor kappa B (NF-κB) activities; suppress neutrophil extracellular trap (NET) formation; regulate phagocytosis, cell apoptosis, proliferation, and differentiation; and potentially influence organ fibrosis. Administration of recombinant SLPI has been shown to ameliorate kidney dysfunction, reduce tubular necrosis, and attenuate kidney inflammation in IR and nephrotoxin-induced AKI models [18]. Furthermore, SLPI conferred protection in cisplatin-induced AKI by suppressing NF-κB signaling in macrophages [19]. This review article assesses evidence for the roles and mechanisms of SLPI in both tissue injury and repair processes, highlighting its potential as a diagnostic and prognostic biomarker, as well as a therapeutic candidate, to ameliorate AKI and potentially CKD.

## 2. Molecular Structure and Classical Functions of SLPI

SLPI was originally found in mucosal secretions of the respiratory, digestive, and reproductive tracts [10,11,12]. It is produced by inflammatory cells such as macrophages and neutrophils [13,14], as well as mucosal epithelial cells [11]. Mature human SLPI is an approximately 11.7 kDa nonglycosylated cationic protein of 107 amino acids that is derived from a 132 amino acid precursor after cleavage of a 25 amino acid signal peptide [20]. It is a single-chain polypeptide that belongs to the whey acidic protein family [21]. It possesses two homologous structural domains: the N-terminal domain (residues 1–54) and the C-terminal domain (residues 55–107), a feature conserved in porcine and murine SLPI [22]. The protease inhibitory activity of SLPI resides in the C-terminal, where the Leu72-Met73-Leu74 motif serves as the binding site for serine proteases, notably neutrophil elastase (NE), trypsin, and cathepsin G [20,23]. This antiprotease activity protects the epithelia against excessive proteolysis at sites of inflammation characterized by abundant infiltrated immune cells. In addition, both the antibacterial and antifungal activity of SLPI reside primarily in its N-terminal domain, which lacks protease inhibitory activity [24]. This property is attributed to the high positive charge of the N-terminal domain, which interacts with anionic bacterial lipids, thereby leading to disruption of the bacterial cell wall [25,26]. SLPI also demonstrates antiviral activity by interfering with the entry of human immunodeficiency virus type 1 (HIV-1) into target cells such as macrophages. It can competitively bind to annexin II or the phospholipid scramblase 1 (PLSCR1), proteins considered essential for mediating the intracellular transport of HIV-1 [27,28,29]. Although the antiprotease and antimicrobial activities of SLPI constitute its first line defense for the host, the full spectrum of biological functions of SLPI extends beyond these roles and remains incompletely understood.

## 3. Regulation of SLPI Expression

At the genome level, the impact of DNA methylation has been implied in the progression of renal disease [30]. The transcription of the SLPI gene is epigenetically regulated by DNA methylation, a key mammalian mechanism involving the addition of a methyl group at the C5 position of cytosine within CpG sites or islands. Methylated CpG can recruit methyl-CpG binding protein 2 (Mecp2) to the nucleus, where it binds methylated cytosines and represses gene transcription [31]. In HEK293T, a human embryonic kidney cell line, a plasmid containing the SLPI promoter was co-transfected with increasing concentrations of plasmids expressing Mecp2 [32]. SLPI protein expression was progressively downregulated with higher amounts of Mecp2 plasmid. This suggests that Mecp2 binds methylated CpG sites in the SLPI promoter and negatively regulates its expression. Chromatin immunoprecipitation confirmed Mecp2 binding to the SLPI promotor region. Collectively, these data indicate that SLPI expression is regulated through promoter methylation and subsequent Mecp2 recruitment. However, the pathophysiological relevance of this regulatory mechanism in human kidney diseases remains unclear.

Beyond epigenetic regulation, SLPI is modulated by its protease substrates. In primary human airway epithelial cells, exogenous NE significantly upregulates SLPI mRNA levels, whereas other neutrophil products, including cathepsin G, myeloperoxidase, and lysozyme, do not [33]. Non-neutrophil proteases like trypsin and pancreatic elastase similarly increase SLPI transcription. Paradoxically, in cell-free systems, NE and cathepsin G directly degrade SLPI protein, reducing its concentration in supernatants [34]. This suggests dual-layer regulation by proteases: transcriptional induction versus post-translational degradation, potentially maintaining SLPI homeostasis. Cell type-specific differences govern the regulatory interplay between NE and SLPI. In the respiratory tract epithelium, NE induces SLPI gene transcription while simultaneously degrading SLPI protein, a dual-layer mechanism proposed to form a feedback loop that dynamically balances their mutual levels. Conversely, in lung carcinoma contexts (e.g., A549 cells), NE suppresses SLPI secretion, an effect linked to its tumor-promoting function [34,35]. Myeloid cells exhibit a third pattern: exogenous NE upregulates SLPI at both transcriptional and translational levels, with NE inhibition conversely reducing SLPI expression [36]. Collectively, these observations demonstrate that NE regulates SLPI through compartment-specific mechanisms—spanning transcriptional induction, post-translational degradation, and secretory modulation—to drive divergent biological outcomes across physiological and pathological milieus.

In addition, Vandooren and colleagues reported that matrix metalloproteinase (MMP)-9 and MMP-8 cleave SLPI predominant at the C-terminus at a low MMP:SLPI ratio [37]. In contrast, high concentrations of these MMPs cleave SLPI at both N- and the C-termini. MMP-2 and MMP-7 exhibited similar proteolytic activity against SLPI, whereas MMP-3 did not. Defensins—small cationic proteins involved in innate immunity—enhance SLPI secretion without altering its mRNA levels in primary bronchial epithelial cells [38], suggesting pre-formed SLPI is stored intracellularly for rapid release during inflammatory insults. Notably, productive HIV-1 infection is dispensable for SLPI induction. Both heat-inactivated and infectious viral particles upregulate SLPI transcription and protein secretion in human oral keratinocytes [39]. Beyond epithelial cells, macrophages contribute critically to innate immune regulation via SLPI secretion. Mycobacterium tuberculosis activates TLR2 signaling in macrophages, enhancing SLPI expression and secretion [40]. Collectively, these studies demonstrate the role of SLPI in innate immune responses, with its expression regulated through transcriptional, post-translational, and secretory mechanisms depending on cellular context.

## 4. SLPI as a Potential Biomarker in Acute and Chronic Kidney Diseases

### 4.1. SLPI as a Promising Early Diagnostic Biomarker for AKI Post-Cardiovascular Surgery

AKI is one of the most common complications following cardiovascular surgery and is closely associated with increased mortality. Surgical procedures involving aortic cross-clamping and cardiopulmonary bypass (CPB) frequently induce renal hypoperfusion by low-flow and subsequent IR injury upon flow restoration. CPB and surgical injury to tissues can also induce the activation of systemic inflammation and oxidative stress, leading to tubular cell injury [41]. Standardized surgical procedures such as CPB provide precisely timed IR injury windows, enabling dynamic biomarker monitoring. A prospective observational study evaluated serum and urine SLPI levels in patients following cardiac surgery [15]. Patients who developed AKI of all stages (I-III), as defined by KDIGO criteria, exhibited significantly elevated serum SLPI levels as early as 6 h post-operation, peaking at 24 h, compared to non-AKI patients. In contrast, urinary SLPI levels did not differ significantly between the two groups, suggesting that serum SLPI may be more useful than urinary SLPI for predicting AKI. Another study focused on thoracoabdominal aortic aneurysm (TAAA) repair, a procedure in which AKI is a common postoperative complication associated with patient outcomes [42]. Patients who developed AKI showed higher serum SLPI levels at both 12 h and 24 h after admission to the ICU compared to those without AKI [43]. At the 12 h time point, SLPI demonstrated a sensitivity of 76.47% and specificity of 87.5% for predicting AKI. Furthermore, since no significant association was observed between perioperative SLPI levels and postoperative sepsis, mortality, major cardiovascular events, or pneumonia, serum SLPI appears to serve as a specific biomarker for AKI following TAAA repair (summarized in Figure 1). Although these studies support the diagnostic potential of SLPI, further analyses of longitudinal changes in serum SLPI within the AKI group could improve individual risk stratification.

### 4.2. SLPI Predicts Graft Outcomes Post-Kidney Transplantation

Kidney transplantation, the optimal treatment for end-stage renal failure, often involves IR injury, leading to AKI, and impairs graft function [44]. Studies have shown that elevated SLPI levels in the cold preservation solutions were associated with delayed graft function or rejection episodes post-transplantation [45]. Thus, SLPI concentration in perfusion solutions may help assess donor kidney quality and predict early allograft function. One study reported significantly higher SLPI mRNA levels in biopsies from grafts with AKI compared to controls immediately after transplantation [46]. Similarly, serum SLPI was elevated in the AKI group on the day of transplantation. Due to the shortage of living donors, deceased donors represent a major source of transplantable kidneys [47]. However, deceased donors have a significantly lower 10-year survival rate than living donor grafts, posing a challenge for the long-term prognosis [48]. A whole-transcriptome analysis of biopsy samples from human kidney transplants revealed that SLPI was significantly upregulated by 5.7-fold at 30 min and 3.2-fold at 3 months in deceased donors compared to living donors [16]. At 12 months post-transplantation, deceased donors developed notable tubulointerstitial fibrosis, which was absent in living donors. These findings suggest that SLPI in serum or graft biopsies may serve as a novel biomarker reflecting acute donor responses and long-term transplant outcomes.

### 4.3. SLPI in Chronic Kidney Disease Progression

Although the role of SLPI in predicting AKI-to-CKD transition remains unexplored, one study measured serum SLPI in patients with type 2 diabetes, a leading cause of CKD and end-stage renal disease. Significantly higher serum SLPI levels were observed in patients with established and advanced diabetic kidney disease compared to healthy controls or those with early-stage disease [49]. Receiver operating characteristic curve analysis indicated that a serum SLPI level above 39.11 ng/mL could predict diabetic kidney disease. During an 80-month follow-up, Kaplan–Meier analysis suggested that levels ≥51.61 ng/mL were associated with an increased risk of end-stage renal disease. Additionally, clinical analyses of renal interstitial fibrosis in CKD patients revealed elevated SLPI mRNA levels in fibrotic areas compared to the healthy control, a finding consistent with observations in unilateral ureteral obstruction mice models [50]. These results imply that SLPI may also serve as a biomarker for chronic kidney injury progression.

### 4.4. Challenges and Specificity

A key consideration for SLPI as a systemic biomarker is tissue specificity. As SLPI is produced at various inflammatory sites, elevated serum levels are not exclusive to renal damage. However, evidence supports the kidney as a significant source of SLPI during injury. Immunohistochemical staining and in situ hybridization confirm localized expression of SLPI in renal tubular cells under both physiological and pathological conditions. Specifically, SLPI is produced in distal tubules, and its mRNA is detectable in human kidney biopsies, indicating de novo synthesis within the kidney rather than mere filtration from circulation [51]. HK-2 culture demonstrated upregulated expression of SLPI protein under hypoxia stimulation [52]. The temporal kinetics and magnitude of the SLPI increase may help distinguish AKI from other inflammatory states. Future studies should evaluate SLPI as part of a multi-marker panel to enhance specificity for renal injury.

In summary, clinical evidence supports the role of SLPI as an alarm protein upregulated in the acute phase of AKI, highlighting its potential for early diagnosis. Serum SLPI levels may also offer prognostic value for CKD, emphasizing its relevance in the AKI to CKD transition. Figure 1 summarizes clinical and experimental evidence supporting the role of SLPI as a biomarker across acute and chronic kidney diseases. However, the functional roles and related mechanisms of this acute-phase-responsive molecule in AKI require further investigation.

## 5. SLPI Involvements in Multiple Biological Events Related to AKI

### 5.1. SLPI and Immune Regulation

The anti-inflammatory functions of SLPI extend beyond its direct antiprotease and antimicrobial activities. SLPI inhibited NF-κB transcriptional activity, suppressed pro-inflammatory cytokine production, and blocked NET formation. SLPI exerted organ-protective effects in the lung, liver, and central nervous system through inflammation suppression [53,54,55]. Lipopolysaccharide (LPS) or renal IR injury activated inflammatory responses by binding toll-like receptor 4 (TLR4), and triggering NF-κB signaling and downstream immune cascades [56,57]. NF-κB activation promotes pro-inflammatory cytokine production, mitochondrial dysfunction, tubular cell apoptosis, and macrophage infiltration [58,59]. SLPI-knockout mice exhibited increased mortality versus wild-type controls following LPS challenges, confirming the protective role of SLPI against LPS-induced inflammation [60]. In rodent LPS or renal IR-induced AKI models, exogenous SLPI inhibited NF-κB activation and maintained an anti-inflammatory phenotype of macrophages [19]. Mechanistically, exogeneous SLPI entered and rapidly localized to the cytoplasm and nucleus of U937 monocytic cell line and blocked LPS-induced degradation of inhibitor of NF-κB alpha (IκBα). Consequently, this prevented the release of NF-κB from IκBα and its subsequent nuclear translocation [61,62]. Furthermore, nuclear-localized SLPI competitively bound to the NF-κB DNA-binding site, including promoters of pro-inflammatory cytokines tumor necrosis factor-alpha (TNF-α) and IL-8, thereby suppressed their transcription [63,64]. Studies also showed SLPI attenuated TLR2/4 signaling by reducing monocyte production of interleukin IL-6 and high-mobility group box-1 (HMGB-1) [65,66]. The inhibitory roles of SLPI on NF-κB activation and pro-inflammatory cytokine production imply significance in relation to alleviating AKI. However, the immunoregulatory functions of SLPI in tubular cells and their crosstalk with immune cells during AKI pathogenesis remain poorly characterized.

NETs—composed of DNA, histones and cytoplasmic/granular proteins—drive persistent tissue damage and inflammation in AKI [67]. NETs promoted tubular necrosis and its induced necroinflammation [67,68]. Exogenous wild-type SLPI reduced TNF-α-induced NET formation by about 60% in neutrophils, an outcome which was correlated with inhibition of histone H4 cleavage [69,70]. In neutrophil nuclei, SLPI was found to be co-localized with NE, which mediated histone proteolysis that contributed to chromatin decondensation and NET formation [71]. Evidence proved that a C-terminal mutant SLPI (impaired in protease inhibition) still retained partial NET-suppressive activity with histone H4 cleavage [69]. Thus, the NET-suppressive activity of SLPI mutants demonstrates that this function operates independently of antiprotease activity, likely through direct histone cleavage. This complements the canonical NE antagonism of SLPI to achieve maximal NET formation control, with broad implications for inflammatory disorders.

Thus, SLPI mediated its anti-inflammatory effect mainly via suppressing NF-κB signaling and NET formation. Further elucidation of this immunomodulatory mechanism may inform novel therapeutic strategies for inflammatory conditions.

### 5.2. SLPI and Phagocytosis

Phagocytosis is an essential process for the clearance of not only “non-self” materials but also “self” components such as cellular debris, playing a critical role in modulating immune responses and initiating tissue repair. This process requires the recognition of dying cells by phagocytes through a complex network involving “find-me”, “eat-me”, and “don’t eat-me” signals, bridging molecules, and specialized phagocytic receptors [72,73]. In the injured kidney, proximal tubular epithelial cells have been identified as the primary phagocytes [74]. Under IR or cisplatin stimuli, these cells rapidly upregulated the expression of KIM-1 on their surface, which served as a receptor for phosphatidylserine (PS) exposed on the outer membrane of apoptotic cells [75]. This “eat-me” signal was enhanced by opsonins such as properdin, which bridged PS and phagocytic receptors on macrophages [76]. Furthermore, genetic knockout of properdin in proximal tubular epithelial cells (semi-professional phagocytes) impaired their phagocytic capacity of FITC-labeled *Escherichia coli* conjugates [77]. However, the precise mechanisms governing phagocytosis, including the roles of signal mediators, phagocytic receptors, and their regulatory pathways during AKI, are still under investigation.

Recent studies have shown that SLPI can function as a pattern-recognition molecule that assists in identifying “non-self” targets, mediates phagocytic functions, and regulates subsequent immune responses. On the cellular membrane, SLPI interacted with annexin II (also known as annexin A2), a protein that bound PS and was involved in phagocytic uptake [27]. This interaction inhibited HIV-1 entry into macrophages, indicating a potential negative regulatory role in phagocytosis that may protect against viral infection; in tuberculosis patients, endogenous SLPI bound to mannans, a pathogen-associated molecular pattern; and on mycobacteria, facilitating their uptake by macrophages, suggesting an opsonizing role for SLPI [78]. Studies have shown that soluble pattern-recognition receptors, such as Pentraxin-3 and Collectins (e.g., SP-A and SP-D), employ a conserved mechanism to opsonize both bacteria and apoptotic cells [79,80,81]. We hypothesize that SLPI, as an opsonin, could extend to the opsonization of apoptotic cells, a critical process for the non-inflammatory clearance of cellular debris. Analogous to other pattern-recognition molecules like Pentraxin-3, the facilitation of apoptotic cell phagocytosis by SLPI would represent a potent mechanism to limit necrosis and promote tissue repair in the injured kidney. In contrast, treatment of human monocytes with recombinant SLPI did not affect their phagocytosis of FITC-labeled *E. coli* [54], but it enhanced their uptake of apoptotic neutrophils [82]. Such evidence indicated that SLPI may have opsonization roles that vary from microbe types. The exact mechanism that SLPI incubation enhanced the uptake of apoptotic cells by monocytes is still unclear. Notably, SLPI expression can also be induced following phagocytosis. Uptake of apoptotic cells by macrophages stimulated SLPI production and secretion [13]. Even mere attachment of apoptotic cells to macrophages, without full engulfment, can upregulate SLPI expression. This increase in SLPI subsequently suppressed TNF-α expression, potentially helping to resolve inflammation during the clearance of apoptotic cells.

Thus, SLPI appeared to function as an opsonin for bacteria, an inhibitor/enhancer of phagocytic activity, and a modulator of immune responses post-phagocytosis. Further investigation is warranted to determine whether SLPI can opsonize dead cells or cellular debris, facilitate their recognition by professional (e.g., macrophages) or semi-professional phagocytes (e.g., kidney tubular cells), and regulate phagocytic function across different cell types. By enhancing opsonization and clearance of apoptotic cells, a process critical for resolving inflammation and initiating tissue repair, SLPI directly links its immunomodulatory functions to the promotion of renal recovery.

### 5.3. SLPI and Cell Apoptosis, Proliferation, and Differentiation

It has been noted that SLPI can determine the fate of diverse cell types. SLPI was reported to promote growth, proliferation, and differentiation of endometrial epithelial cells, osteoblasts, and neural cells [83,84,85]. Mechanistically, SLPI upregulated Osterix (essential for osteoblast differentiation/proliferation), enhanced cyclin D1 (cell-cycle promotor), induced HES1 (basic helix–loop–helix transcription factor regulating differentiation), and suppressed insulin-like growth factor-binding protein-3 (anti-proliferative and anti-apoptosis factor). In the human kidney proximal tubular cell line HK-2, SLPI incubation enhanced viability and inhibited apoptosis during serum deprivation or nephrotoxin gentamicin exposure. It also increased proliferation of both HK-2 cells and mouse proximal tubular cells [18,52]. Overexpression of SLPI in proximal tubular cells markedly enhanced the abundance of cyclin D1 and Ki-67 for regeneration. It is suggested that SLPI can coordinate tubular epithelial cell survival and proliferation to exert nephroprotective and reparative functions in AKI.

Moreover, SLPI modulated the survival and proliferation of immune cells that play an essential role in the development and progression of AKI. During the repair stage, infiltrated monocytes differentiate into pro-inflammatory M1 macrophages that subsequently transition to an anti-inflammatory M2 phenotype. While M2 macrophages promote tubular regeneration, their prolonged persistence may drive renal fibrogenesis [86]. Notably, SLPI inhibited TNF-α-induced apoptosis in peripheral blood monocytes by suppressing caspase-3 activation, an effect maintained even by antiprotease-deficient SLPI variants, confirming its non-proteolytic antiapoptotic function [87]. Similarly, recombinant SLPI reduced TNF-α-induced neutrophil apoptosis by 73% [88]. These robust anti-apoptotic effects on monocytes and neutrophils likely contribute to the persistent accumulation of immune cells (including M2 macrophages) during later AKI phases, potentially exacerbating tissue injury and compromising renal repair processes. SLPI also exerted critical immunomodulatory effects on adaptive immunity. B cells have been shown to promote tubular atrophy and fibrosis in renal IR and unilateral ureteral obstruction models by impairing tubular regeneration and repair during the recovery phase [89,90]. SLPI deficiency was found to enhance B-cell proliferation and IgM production in response to LPS stimulation [66]. This regulatory function may protect against B cell-mediated impairment of tubular regeneration during AKI recovery. Persistent CD4^+^ T-cell activation during post-IR recovery phases can induce delayed albuminuria, as demonstrated by adoptive transfer experiment [91,92]. This evidence established CD4^+^ T cells as a key mediator in the progression of AKI. Mechanistically, SLPI orchestrated a multi-tiered immunomodulatory program through monocyte–T-cell crosstalk. Soluble factors, mostly IL-4 from SLPI-primed monocytes, directly inhibit CD4^+^ T-cell proliferation [93]. SLPI-modified monocytes also selectively suppressed T-lymphocyte activation and differentiation into pro-inflammatory T helper type 1 (Th1, CD4^+^ T subtype) cells, while enhancing anti-inflammatory Th2 cell production of IL-4 and IL-10 in vitro [93].

Overall, the above evidence suggests that SLPI may contribute to the renal recovery through tissue-protective and immunomodulatory mechanisms. By enhancing tubular epithelial cell survival and proliferation, it could facilitate the structural restoration of injured kidneys. SLPI’s pleiotropic immunoregulatory effects, encompassing inhibition of lymphocyte proliferation (B cells and CD4^+^ T cells), preferential attenuation of Th1 activity, and enhancement of Th2 responses, may synergistically favor the establishment of a pro-regenerative immunological microenvironment. Notably, SLPI-mediated prolonged survival of monocytes and neutrophils may paradoxically fuel fibrosis, highlighting its context-dependent therapeutic window for AKI intervention.

### 5.4. SLPI and Fibrosis

The transition from AKI to CKD is a clinically significant process marked by progressive fibrosis. This progression has been extensively studied characterized in animal models induced by renal IR injury or nephrotoxic agents, consistently demonstrating tubular atrophy and interstitial fibrosis. Multiple cell types, including fibroblasts, macrophages, and tubular epithelial cells, along with their interactions, are known to play essential roles in driving fibrotic processes [94,95]. Matrix metalloproteinases have been implicated in the pathogenesis of kidney fibrosis through their regulation of extracellular matrix remodeling [96]. MMP-9, in particular, was upregulated by the profibrotic cytokine TGF-β1 in macrophages [97]. Its abundant expression promoted disruption of E-cadherin and facilitated epithelial–mesenchymal transition in tubular cells, ultimately driving renal fibrosis [98]. Notably, MMP-9 can proteolytically cleave SLPI at both functional domains [37]. Supporting evidence from patients with non-cystic fibrosis bronchiectasis revealed the presence of SLPI cleavage fragments in lower airway secretions, alongside high MMP-9 levels. Studies also indicated that SLPI negatively regulated MMP-9 expression in monocytes and resident peritoneal macrophages by inhibiting prostaglandin H synthase-2 or mitogen-activated protein kinases signaling [99]. In contrast, TGF-β1 has been reported to downregulate SLPI expression in bronchial epithelial cells under hypoxia conditions [100].

Fibroblasts not only produce excessive collagen but also reorganize and compact the collagen fibers, leading to progressive stiffening of the extracellular matrix and scar formation [101]. The collagen gel contraction assay provided an in vitro three-dimensional model to study mechanical interactions between cells and the extracellular matrix [102]. Evidence indicated that SLPI secreted by oral epithelial cells can inhibit fibroblast-mediated collagen gel contraction [103]. Specifically, SLPI in culture medium was found to alter fibroblast morphology by inhibiting cytoskeletal development and increasing cyclic AMP levels, which may influence collagen gel contraction. Exogeneous SLPI treatment reversed transforming growth factor β1 (TGF-β1)-induced excessive gel contraction by fibroblasts. Furthermore, plasmid-mediated overexpression of SLPI in fibroblasts reduced their gel-contracting activity, an effect reversed by anti-SLPI antibody treatment.

These findings suggest that exogenous SLPI supplement may improve long-term outcomes after kidney injury by attenuating fibrosis through modulation of suppression of MMP-9 and collagen gel contraction.

## 6. Therapeutic Potential of SLPI in AKI

### 6.1. SLPI Alleviates Kidney Damage in Experimental AKI Models

Our previous study demonstrated that the erythropoietin-derived cyclic helix B surface peptide (CHBP) and small interfering RNA (siRNA) targeting caspase-3—an apoptosis execution protein— exerted renoprotective effects in a mouse model of kidney IR-AKI. Microarray analysis revealed that SLPI was significantly upregulated by caspase-3 siRNA in CHBP-treated kidneys following IR 48 h [104]. The renoprotective efficacy of SLPI has been demonstrated in multiple well-controlled in vivo and in vitro models of AKI (summarized in Table 1). In a murine kidney IR model, intraperitoneal administration of exogenous recombinant human SLPI significantly reduced elevated plasma creatinine and blood urea nitrogen levels, attenuated tubular epithelial cell death, and decreased neutrophil infiltration at 24 h post-IR compared to untreated IR animals. The renoprotective effect of SLPI was also confirmed by preserving the renal function in gentamicin-induced AKI rats compared to the untreated animals at 96 h. In vitro, SLPI incubation enhanced cell survival in HK-2 during serum starvation or FK506 exposure than control cells. It also increased proliferation and migration of HK-2 cells [18]. SLPI protein was found to have upregulated expression in HK-2 under hypoxia stimulation. Overexpression of SLPI by plasmid vectors in proximal tubular cells markedly enhanced the abundance of cyclin D1 and Ki-67 controlled by the empty plasmid [52]. Transcript analysis showed that SLPI treatment downregulated the expression of several monocyte-related markers elevated by IR injury, including CD86, CD68, and CD14; the monocyte chemokine CCL2; and the pro-inflammatory cytokine TNF-α. Interestingly, SLPI also reduced the transcript level of the anti-inflammatory cytokine IL-10, which may reflect overall attenuation of inflammation [18]. Furthermore, in HK-2 cells, SLPI promoted cell proliferation and migration, suggesting a potential role in facilitating kidney repair, which needs further validation.

In a cisplatin-induced mouse AKI model, isorhamnetin alleviated renal injury, concomitant with upregulation of SLPI and downregulation of pro-inflammatory cytokines such as IL-1β, IL-6, and TNF-α at 72 h [19]. Isorhamnetin suppressed the macrophage inducible C-type lectin/spleen tyrosine kinase/NF-κB pathway via a SLPI-dependent manner in LPS-stimulated RAW264.7 macrophages. In vitro study proved that SLPI knockdown in RAW264.7 promoted activation of this pathway and enhanced expression of M1 phenotype-associated cytokines. It demonstrated that SLPI served as an essential mediator of the renoprotective role of cisplatin-induced AKI.

In summary, SLPI exerts renoprotective effects in AKI by attenuating inflammation, e.g., by suppressing CCL2 and TNF-α, reducing neutrophil infiltration, and promoting tubular epithelial cell repair, potentially via modulating macrophage polarization of inhibiting M1 phenotype and the NF-κB pathway. Future studies should elucidate the dynamic role of SLPI across AKI stages, particularly its crosstalk with cell survival and immune regulation, and explore SLPI-directed therapeutic strategies such as combination drugs or gene-based approaches to improve renal recovery.

### 6.2. Safety Considerations and Challenges for Clinical Translation

Before proposing clinical trials, a thorough discussion of safety is paramount. SLPI is an endogenous protein with a generally anti-inflammatory profile by inhibiting the release of pro-inflammatory cytokines, NET formation, and leukocyte infiltration (summarized in Figure 2, left penal). However, In vitro studies also showed SLPI prevented the apoptosis of neutrophils and macrophages, thus indicating the persistent existence of these inflammatory cells. Therefore, prolonged administration of SLPI might theoretically impede necessary inflammation or sustain chronic inflammation and subsequent tissue fibrotic damage. Additionally, the clinical translation of SLPI in oncology requires careful safety considerations due to its dual role in inflammation and cancer. While SLPI exerts potent anti-inflammatory and tissue-protective effects, its overexpression in multiple carcinomas is associated with pro-tumorigenic effects, potentially through suppression of anti-tumor immunity and promotion of metastasis [105]. To address the safety concerns of SLPI clinical translation, key strategies would be considered, including timely and temporally intervention during active injury phase, developing localized delivery systems to minimize systemic exposure and engineering functionally selective SLPI variants that retain anti-inflammatory activity but lack pro-tumorigenic effects.

## 7. Conclusions and Perspectives

This review comprehensively examines the emerging roles of SLPI in AKI, highlighting both its diagnostic and therapeutic potential. Clinical evidence demonstrated that SLPI is significantly upregulated during the acute phase of AKI, suggesting its potential as an early diagnostic indicator for renal injury. Importantly, preclinical studies in IR and nephrotoxic AKI models consistently show that SLPI administration ameliorates kidney dysfunction, reduces tubular injury, and attenuates inflammatory responses. The antifibrotic properties of SLPI, particularly its ability to modulate collagen gel contraction and suppress fibrotic signaling, further suggest its therapeutic potential in preventing AKI progression to chronic kidney disease. Accumulating evidence highlights the involvement of SLPI in multiple biological processes associated with AKI, including the regulation of immune responses, phagocytosis, cell survival, proliferation, and differentiation, as well as the progression to CKD and fibrosis (summarized in Figure 2). The role of SLPI in mediating or regulating the phagocytic function of renal tubular epithelial cells (semi-professional phagocytes) remains unclear, despite its critical involvement in post-AKI repair. Further investigation is needed to elucidate whether SLPI directly enhances tubular phagocytosis, which could unveil novel therapeutic strategies for mitigating kidney injury and fibrosis. This expanding understanding of pleiotropic actions of SLPI not only deepens our knowledge of renal repair mechanisms but also provides a strong foundation for developing novel interventions against AKI and its transition to CKD.

Future research in several key directions warrants in-depth investigation. First, clinical validation studies are needed to further determine its diagnostic value in diverse AKI populations. Second, the functional domains of SLPI on renoprotection still remains elusive, thus requiring us to optimize or overcome challenges such as protease susceptibility by differentiating the diverse roles of different domains of SLPI. Third, elucidating tissue-specific regulatory mechanisms of SLPI will be crucial for understanding its context-dependent effects in different AKI etiologies. Finally, exploring the crosstalk between SLPI and other inflammatory pathways may uncover novel combination therapies for AKI and CKD. By addressing these questions, future research could translate the promising preclinical findings of SLPI into clinically viable strategies for early diagnosis and prognosis, as well as precision treatment of AKI.

This integrated perspective not only advances our understanding of the multifaceted roles of SLPI but also provides a roadmap for developing innovative approaches to AKI management, bridging the gap between basic research and clinical application.

## Figures and Tables

**Figure 1 ijms-26-11631-f001:**
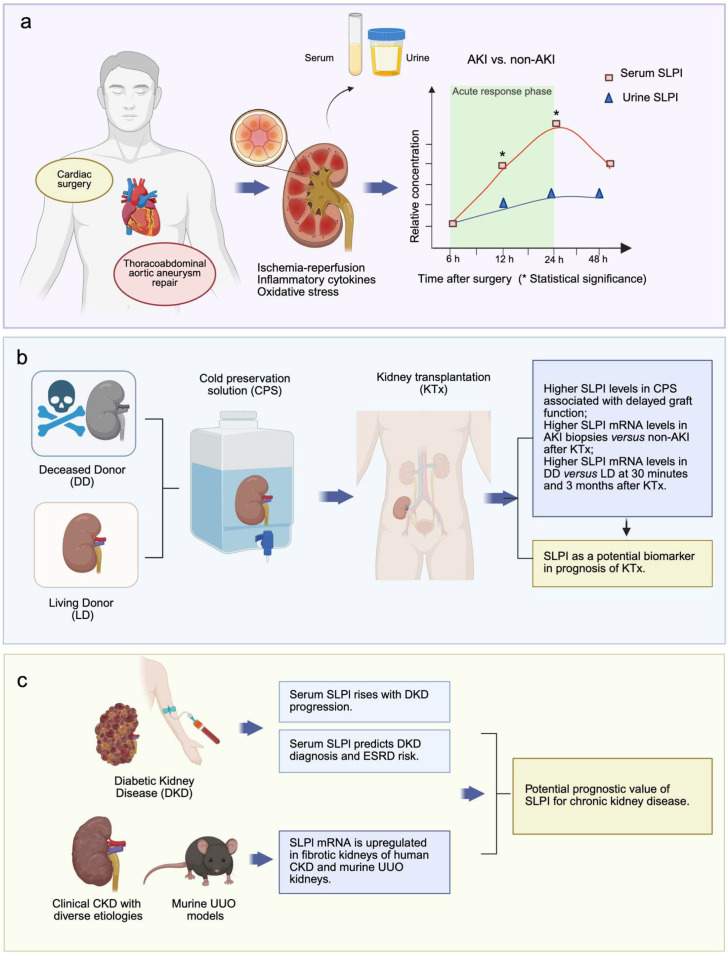
SLPI as a potential biomarker for acute and chronic kidney disease. (**a**) Cardiac and thoracoabdominal aortic aneurysm repair surgeries can induce AKI to patients by exposing the kidney to ischemia–reperfusion injury, inflammatory cytokines, and oxidative stress. These pathological processes trigger the release of SLPI into the circulation and urine. The plotted curves illustrate differences in serum or urinary SLPI levels between AKI and non-AKI patients during the first 48 h post-cardiovascular surgery. Patients who developed AKI exhibited significantly higher serum SLPI levels than those in non-AKI patients at 12 h and 24 h after surgery during the acute responding phase [15,43]. In contrast, urinary SLPI shows a slower and less pronounced increase. Serum SLPI may serve as a superior biomarker to urinary SLPI for AKI prediction. (**b**) Findings link elevated SLPI levels in preservation solutions and graft biopsies to delayed graft function, AKI, and inferior outcomes in DD compared to LD grafts. (**c**) Evidence demonstrates that serum SLPI levels increase with DKD progression and can predict diagnosis and end-stage renal disease risk. Furthermore, SLPI mRNA is upregulated in fibrotic kidneys from patients with diverse CKD etiologies and in murine UUO models, indicating an association with renal fibrosis. Collectively, these data position SLPI as a promising biomarker for early diagnosis and prognosis of AKI, and disease progression in chronic kidney disease. AKI, acute kidney injury; CKD, chronic kidney disease; DD, deceased donor; DKD, diabetic kidney disease; LD, living donor; SLPI, secretory leukocyte protease inhibitor; UUO, unilateral ureteral obstruction. Created in BioRender. Rui, C. (2025) https://BioRender.com/p2i4tfm, accessed on 30 November 2025.

**Figure 2 ijms-26-11631-f002:**
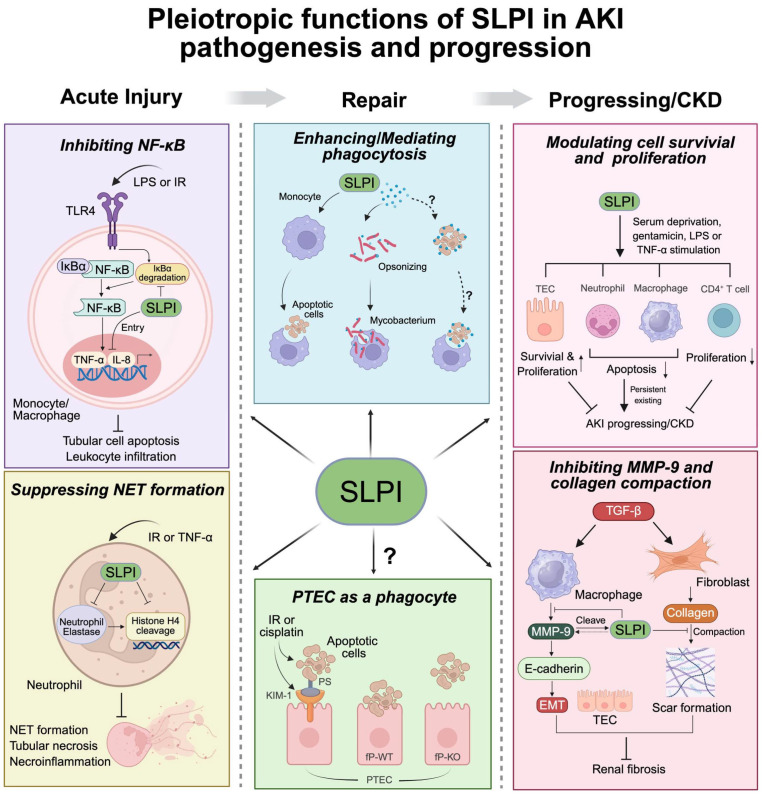
Pleiotropic roles of SLPI in the pathogenesis and progression of AKI. The multifaceted roles of SLPI, encompassing immunomodulation, regulation of cell survival/proliferation, phagocytosis, and antifibrotic activity, collectively establish a dynamic network that orchestrates renal protection during AKI. Therapeutic administration of SLPI demonstrates protective and reparative effects in preclinical models. Left penal: In the acute injury phase, SLPI attenuates inflammation by inhibiting nuclear factor kappa B (NF-κB) activation through the prevention of inhibitor of nuclear factor kappa B alpha (IκBα) degradation in the cytoplasm [61,62]. SLPI also translocates to the nucleus and binds directly to the promoters of tumor necrosis factor-α (TNF-α) and interleukin-8 (IL-8), suppressing their transcription. Concurrently, SLPI inhibits neutrophil elastase activity and the cleavage of Histone H4, thereby suppressing the neutrophil extracellular trap (NET) formation [63,64]. Middle panel: During the repair phase, SLPI treatment enhances monocytes to uptake apoptotic cells [82]. SLPI also mediates phagocytic clearance of pathogens by monocytes via opsonization [78]. It is still unknown whether SLPI also opsonizes apoptotic cells and mediates their phagocytosis. Upon injury, the proximal tubular epithelial cell (PTEC) expresses kidney injury molecule-1 (KIM-1) to mediate their uptake of apoptotic cells via recognizing phosphatidylserine (PS) [75]. The complement factor properdin (fP) is essential for maintaining the phagocytic function of PTEC [77]. However, it is unknown whether SLPI can regulate the phagocytosis of the PTEC. Right panel: In the progression to CKD or fibrosis, SLPI differentially modulates the survival and proliferation of the tubular epithelial cell (TEC) [18,52], macrophage [87], neutrophil [88], and lymphocytes [66,93], as indicated. In addition, it inhibits matrix metalloproteinase-9 (MMP-9) release by the macrophage and cleaves MMP-9; it also inhibits collagen compaction during scar formation [99,103]. EMT, epithelial–mesenchymal transition; IR: ischemia–reperfusion; KO, knockout; LPS, lipopolysaccharide; TGF-β, transforming growth factor-β; WT, wild type. Created in BioRender. Rui, C. (2025) https://BioRender.com/94efb3v, accessed on 30 November 2025.

**Table 1 ijms-26-11631-t001:** The effects of SLPI on AKI in in vivo and in vitro studies.

Animal/Cell Models	Intervention	Readout	Outcomes	References
Male Wistar rats, bilateral kidney ischemia 40 min, reperfusion 24 h	SLPI 250 μg/kg, i.p., given at one dose on the time of reperfusion or three doses at 24 h pre-ischemia, during the ischemia, and 6 h post-ischemia	Reduction in SCr, BUN, acute tubular necrosis, MPO activity, mRNA expression of CD86, CD68, CD14, CCL2, TNF-α, IL-10, and C3αR	SLPI preserved kidney function and structure, ameliorated inflammation	[18]
Male Wistar rats, gentamicin-induced AKI	Gentamicin 60 mg/kg/day or plus SLPI 250 μg/kg, i.p., once daily for 4 days	Reduction in SCr, BUN	SLPI preserved kidney function	
Human proximal tubule cell line HK-2, serum starvation, treatment with immunosuppressant FK506, or overexpression of SLPI	Serum deprivation, or plus SLPI 0.4 μg/mL for 24 h; FK506 10 μM or plus SLPI 0.4 or 4 μg/mL for 24 h; overexpression of SLPI by transfection of plasmid vectors, controlled by empty plasmids	Reduction of cell apoptosis and necrosis; increased cell counting; increased cell-covered area in wound-healing scratch assay; increased Cyclin D1 and Ki67 staining	SLPI improved cell survival and proliferation	[18,52]
Adult male C57BL/6 mice, cisplatin-induced AKI	Cisplatin 20 mg/kg, i.p., single injection, or plus isorhamnetin 50 mg/kg, i.p., once daily for 3 days	Cisplatin reduced SLPI expression; isorhamnetin increased SLPI and reduced SCr, BUN, and tubular damage	SLPI mediated renoprotection of cisplatin-induced AKI by inhibiting Mincle/Syk/NF-κB-related inflammation in macrophages	[19]
LPS-stimulated macrophage cell line RAW264.7	LPS 300 ng/mL, isorhamnetin 20 μM, or plus SLPI siRNA transfection	Knockdown of SLPI increased expression of Mincle, P-syk, P-P65, iNOS, TNF-α and IL-1β proteins, and IL-1β and IL-6 mRNAs		

## Data Availability

No new data were created or analyzed in this study. Data sharing is not applicable to this article.

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
