# Peer review of "Secretory Leukocyte Protease Inhibitor: A Pleiotropic Molecule for the Potential Diagnosis of and Therapy for Acute Kidney Injury"

_ijms, 2025, doi:10.3390/ijms262311631_

Round 1

Reviewer 1 Report

Comments and Suggestions for Authors

see attached document

Reviewer 2 Report

Comments and Suggestions for Authors

General remarks

  1. The Review summarize the research about the SLPI done for three purposes in AKI and AKI to CKD settings: (1) SLPI as biomarker (BM), (2) SLPI physiological and pathophysiological functional role in pathophysiology of AKI and AKI to CKD conversion and (3) SLPI as interventional agent in AKI development, AKI repair and AKI to CKD conversion. The setting is addressed separately in the manuscript whereas the second and the third are not.
  2. The proposal of the authors to enter clinical phase (line 458-459) is acknowledged but seems as little premature as the safety is not discussed in the manuscript at all. Thus, the structure of the manuscript should be re-arranged.

Specific remarks

  1. SLPI as biomarker.
    1. The purpose of the proposed SLPI as BM is not clearly summarized. It is proposed as both early diagnostic BM (line 16), predictive BM (line 70) and prognostic MB (line 145). As far as this BM is intended for AKI/AKI to CKD developments monitoring and drug development it should follow rules settled for these purposes (https://www.fda.gov/regulatory-information/search-fda-guidance-documents/biomarker-qualification-evidentiary-framework). The authors should adjust/correct the text respectively.
    2. The respective Figure 1 is not informative enough in delineating (1) claimed early diagnostic of BM performance in the setting of AKI vs non-AKI and (2) predictive performance of SLPI for AKI development, AKI recovery (repair) with/without (3) for CKD development, if any. One of the references (#44) failed to show mentioned
  2. SLPI physiological and pathophysiological functional role in pathophysiology of AKI and AKI to CKD conversion. Major part of the manuscript is devoted for this purpose and is clearly written. The Figure 2 is very informative and presents summary of the pathophysiological findings, including remaining uncertainties in clarifying the role of SLPI on phagocytosis. This is welcome. What I would see necessary also – there is a need to clarify the link between the impact on opsonization of bacteria and kidney repair. In the text and in the figure.
  3. SLPI as interventional agent in AKI development, AKI repair and AKI to CKD conversion.
    1. The fundamental gap of the manuscript is the omission of the concept that intervention of the agent should be shown in active controlled setting. For all three purposes (1) AKI development (halting, as is stated in line 26), AKI repair and AKI to CKD conversion.
      1. The both two evidences described in the abstracts available for references #19 and #20 are not informative enough and suggests that there were no controlled arms at all. The authors need to present in the manuscript clear design and results descriptions, preferably together with in separate table   
      2. Before proposing to enter clinical phase, the safety should be discussed in the manuscript as well. In such respect at least the signals of cancerogenicity should be addressed as well (PMID: 33602652).

Reviewer 3 Report

Comments and Suggestions for Authors

The review article by R. Chen et al. offers a comprehensive examination of the emerging role of secretory leukocyte protease inhibitor (SLPI) in acute kidney injury (AKI). This is one of the first reviews to synthesize and critically evaluate the diagnostic and multifaceted therapeutic potential of SLPI specifically during renal damage. The authors consolidate clinical evidence demonstrating SLPI's significant elevation in AKI patients, underscoring its promise as an early diagnostic biomarker. Furthermore, they summarize the pleiotropic biological functions of SLPI, including inhibition of NF-κB, modulation of neutrophil extracellular trap formation, and regulation of apoptosis and fibrosis, which underpin its renoprotective effects. By integrating this evidence, the review successfully proposes SLPI-based strategies as a promising strategy for managing AKI. However, while the diverse roles of SLPI are cataloged, the specific molecular mechanisms and the interplay between its different functions within the kidney are not sufficiently elucidated. A more critical and detailed analysis of these multifaceted roles and the underlying mechanisms would have greatly strengthened the work:

  1. The review currently presents an unclear picture of SLPI's fundamental role in AKI. The abstract and introduction state that serum SLPI levels are elevated in various types of AKI and correlate with poor outcomes. Conversely, the article also presents SLPI as a therapeutic agent with nephroprotective properties, such as reducing inflammation, promoting phagocytosis, and enhancing cell growth and proliferation. In this regard, is SLPI a pathogenic mediator or a protective agent in AKI? If the protein indeed exhibits a context-dependent dual function, this should be explicitly stated and thoroughly discussed in a dedicated section of the review;
  2. Figure 1 currently identifies only cardiac surgery and thoracoabdominal aortic aneurysm as causes for increased serum SLPI. However, the text mentions other clinical conditions, such as kidney transplantation and diabetic nephropathy, which are also associated with elevated SLPI levels. It is recommended to either include these additional pathologies in Figure 1 or discuss them more prominently in the accompanying text to avoid potential misrepresentation of the data;
  3. The review advocates for SLPI as a diagnostic biomarker for AKI, but its tissue specificity is not sufficiently addressed. As the authors note, SLPI is produced by various inflammatory and epithelial cells in multiple tissues. Given that the measured increase occurs in the blood and not specifically in the urine, it is crucial to discuss the evidence that directly links elevated serum SLPI to renal damage, as opposed to a systemic inflammatory response originating from other organs. A more critical analysis of its specificity and universality as a biomarker for kidney injury is needed;
  4. Figure 2 is titled "Pleiotropic roles of SLPI in the pathogenesis and progression of AKI," but it does not differentiate between the potentially beneficial and detrimental effects of the protein. For instance, are anti-inflammatory actions considered protective, while its correlation with poor outcomes indicates a harmful role? To improve conceptual clarity, it would be helpful to visually or descriptively categorize these diverse effects within the figure or its legend;
  5. The inclusion of Mycobacterium in Figure 2, which illustrates the pleiotropic roles of SLPI in AKI, is not substantiated in the text. The connection between SLPI's interaction with Mycobacterium and its direct impact on kidney function or the pathogenesis of AKI is unclear.

Round 2

Reviewer 2 Report

Comments and Suggestions for Authors

Thank you for your amendments and clarifications. The context of SLPI across all three scopes is now sufficiently clear

I have only one remaining issue that was not included in my previous comments (apologies). It concerns Figure 1, where the claimed early diagnostic value of BM performance in the setting of AKI versus non-AKI is presented somewhat confusingly as a comparison between serum and urine. While the different peak levels are important, including a third SLPI curve for the non-AKI case (presumably a flat line) would be very helpful for clarity. Would it be possible to incorporate this adjustment?

Author Response

We appreciate the reviewer's attention to clarity in Figure 1. The curves do not plot raw SLPI levels but represent the calculated difference (ΔSLPI) between AKI and non-AKI patients over time. Therefore, the figure inherently includes non-AKI data as each point on the curve already encodes the relative difference from the non-AKI baseline. We added one sentence ‘The plotted curves illustrate differences in serum or urinary SLPI levels between AKI and non-AKI patients during the first 48 hours post-cardiovascular surgery.’ in line 230-232 of the legend and hope it could enhance clarity.

Additional clarifications:References has been cited in Table 1, which were missing in the previous revised manuscript. 

Reviewer 3 Report

Comments and Suggestions for Authors

I have no further questions. Congratulations on your work.

Author Response

We sincerely thank the reviewer for the previous insightful comments. It helps us to further strengthened the scientific rigor and clarity of our work.